# MiR-21 mediates the protection of kaempferol against hypoxia/reoxygenation-induced cardiomyocyte injury via promoting Notch1/PTEN/AKT signaling pathway

Jinxi Huang[ID]*, Zhenhui Qi

Department of Cardiology, Shanxi Provincial People's Hospital, Taiyuan, Shanxi, P.R. China

* huangjinx01001@qq.com

**Data Availability Statement:** All relevant data are within the manuscript.

**Funding:** The author(s) received no specific funding for this work.

## Abstract

Kaempferol, a natural flavonoid compound, possesses potent myocardial protective property in ischemia/reperfusion (I/R), but the underlying mechanism is not well understood. The present study was aimed to explore whether miR-21 contributes to the cardioprotective effect of kaempferol on hypoxia/reoxygenation (H/R)-induced H9c2 cell injury via regulating Notch/phosphatase and tensin homologue (PTEN)/Akt signaling pathway. Results revealed that kaempferol obviously attenuates H/R-induced the damages of H9c2 cells as evidence by the up-regulation of cell viability, the down-regulation of lactate dehydrogenase (LDH) activity, the reduction of apoptosis rate and pro-apoptotic protein (Bax) expression, and the increases of anti-apoptotic protein (Bcl-2) expression. In addition, kaempferol enhanced miR-21 level in H9c2 cells exposed to H/R, and inhibition of miR-21 induced by transfection with miR-21 inhibitor significantly blocked the protection of kaempferol against H/R-induced H9c2 cell injury. Furthermore, kaempferol eliminated H/R-induced oxidative stress and inflammatory response as illustrated by the decreases in reactive oxygen species generation and malondialdehyde content, the increases in antioxidant enzyme superoxide dismutase and glutathione peroxidase activities, the decreases in pro-inflammatory cytokines interleukin (IL)-1β, IL-8 and tumor necrosis factor-alpha levels, and an increase in anti-inflammatory cytokine IL-10 level, while these effects of kaempferol were all reversed by miR-21 inhibitor. Moreover, results elicited that kaempferol remarkably blocks H/R-induced the down-regulation of Notch1 expression, the up-regulation of PTEN expression, and the reduction of P-Akt/Akt, indicating that kaempferol promotes Notch1/PTEN/AKT signaling pathway, and knockdown of Notch1/PTEN/AKT signaling pathway induced by Notch1 siRNA also abolished the protection of kaempferol against H/R-induced the damage of H9c2 cells. Notably, miR-21 inhibitor alleviated the promotion of kaempferol on Notch/ PTEN/Akt signaling pathways in H9c2 cells exposed to H/R. Taken together, these above findings suggested thatmiR-21 mediates the protection of kaempferol against H/R-induced H9c2 cell injuryvia promoting Notch/PTEN/Akt signaling pathway.

**Competing interests:** The authors have declared that no competing interests exist.

## Introduction

Myocardial ischemia/reperfusion (I/R) injury is a major complication of reperfusion therapy for myocardial infarction, representing a great threat to human health [1, 2]. The underlying molecular mechanisms of myocardial I/R injury are complex, including intracellular calcium overload, acute oxygen stress injury, reduction of nitric oxide bioavailability, mitochondrial damage, and inflammation [3–5]. Although several therapeutic strategies, such as remote ischemic preconditioning and ischemic post-conditioning, have been revealed to attenuate myocardial I/R injury, no effective efficacious therapy for protecting the heart against myocardial I/R exists in clinical practice until now [6, 7]. Hence, it is urgent to discover or develop novel pharmacological agents and therapeutic strategies for I/R-stimulated myocardial damage and the underlying molecular mechanisms.

Kaempferol (3,5,7-trihydroxy-4'-methoxyflavone), a naturally occurring flavonoid, is isolated from the roots of *Alpinia officinarum* which used as a classical tonic agent [8]. Previous studies have revealed that kaempferol has a series of biological activities including anti-inflammatory, anti-oxidant, anti-bacterial, anti-tumor, neuroprotective, and myocardial protective properties [9, 10]. Recent studies find that kaempferol is capable of suppressing myocardial I/R injury [11–13]. The study from Dong Wang *et al.* reveals that kaempferol improves I/R-induced cardiac dysfunction and myocardial injury involved in reducing myocardial infarct size, cardiomyocyte apoptosis, oxidative stress, and inflammatory responses [11]. Another study also shows that kaempferol mitigates myocardial ischemic injury in isolated rat depending on it's antioxidant activity and inhibition of GSK-3β activity [13]. However, little is known regarding the underlying molecular mechanism of action of kaempferol as a therapeutic agent for treating myocardial I/R injury.

MicroRNAs (miRNAs) are endogenous small non-coding RNAs subgroup containing 18–23 nucleotides [14], that functionally modulate specific messenger RNA targets involved in extensive physiological and pathological processes [15, 16]. Increasing evidence reveals that various miRNAs engage in the modulation of cardiovascular formation and function [17, 18], and are the key molecular players in the diagnosis and prevention of myocardial I/R injury [19, 20]. Particularly, miR-21 is highly expressed in cardiomyocytes and has also been shown to regulate important processes such as cardiac fibroblasts, apoptosis, and inflammation in myocardial I/R injury [21–24]. Research confirms that miR-21 expression is significantly reduced after I/R, and miR-21 overexpression reduces myocardial infarct size and cardiomyocytes apoptosis during chronic I/R injury [25, 26]. Recently, miR-21 is also available to involve in a variety of myocardial protective drugs against I/R-induced heart damages *in vivo* and *in vitro* experiments [27–29]. However, the role of miR-21 in the protection of kaempferol on myocardial I/R injury has not been previously reported.

Notch1 signaling pathway is an evolutionarily conserved pathway that is broadly involved in regulating apoptosis, oxidative stress, and inflammatory responses in response to harmful stimuli [30–32]. Previous studies have reported that the Notch1 signaling pathway is also involved in the attenuation of myocardial I/R injury [33, 34]. Research bore out that Nothch1 participates in the inhibitory effect of myocardial protective drugs such as polydatin [35] and Berberine [36] on myocardial I/R injury under diabetic condition by ameliorating oxidative stress damage via activating PTEN/Akt signaling pathway. However, the role of Notch1/ PTEN/Akt signaling pathway in the cardioprotection of kaempferol has not been studied. Notably, research from Cao J *et al.* demonstrates that miR-146a and miR-21 cooperate to accelerate vascular smooth muscle cell proliferation via modulating the Notch signaling pathway in atherosclerosis [37]. Hence, the present study further demonstrated whether Notch1/ PTEN/Akt signaling pathway participates in the contribution of miR-21 to kaempferol-induced cardioprotection against H/R injury.

Here, the results showed that miR-21 mediates kaempferol-induced the inhibition on H/R-induced the damage of cardiomyocytes through reducing oxidative stress and inflammatory response. In addition, miR-21 contributes to kaempferol-led to the enhancement of Notch1/PTEN/Akt signaling pathway, and inhibition of Notch1/PTEN/Akt signaling pathway blocks the cardioprotection of kaempferol under H/R condition, indicating that miR-21 mediates the protective effect of kaempferol on myocardial I/R injury via promoting Notch1/PTEN/AKT signaling pathways *in vitro*. These results potentially provide a new perspective on understanding cardioprotective effect of kaempferol.

## Materials and methods

### Cell culture and H/R treatment

The rat heart-derived cardiac myoblast H9c2 cardiomyocytes were purchased from the American Type Culture Collection (ATCC; Manassas, VA, USA), and were cultured in Dulbecco's modified Eagle's medium (DMEM; Sigma-Aldrich, St Louis, USA) supplemented with 10% (v/v) fetal bovine serum (FBS), 100 U/mL penicillin and 100 μg/mL streptomycin (both from GIBCO, Grand Island, NY, USA) in a humidified incubator with 95% air and 5% $CO_2$ at 37˚C. In order to establish an *in vitro* model of myocardial I/R injury, H9c2 cells were maintained in a hypoxic chamber (95% $N_2$ and 5% $CO_2$) with serum- and glucose-free DMEM at 37˚C for 6 h, to induce hypoxia. After hypoxia, the cells were reoxygenated under normoxic conditions (95% air/5% $CO_2$) with DMEM containing 10% FBS for 12 h at 37˚C [38]. Cells cultured under normal conditions in a humidified incubator with 95% air and 5% $CO_2$ were served as negative controls.

### Experimental group

To determine the protection of kaempferol on H/R injury, cells were randomly allocated to four groups: 1) Control group, no treatment; 2) H/R group, hypoxia (6 h)/reoxygenation (12 h); 3) kaempferol (Kae) + H/R group, pretreatment with different concentrations of kaempferol (5, 10, 20, or 30 $\mu$M) for 2 h prior to H (6 h)/R (12 h); and 4) kae alone group, pretreatment with kaempferol (5, 10, 20, or 30 $\mu$M) for 20 h. To further explore the role of miR-21 in the cardioprotection of kaempferol, cells were randomly allocated to six groups: 1) Control group, no treatment; 2) H/R group, H(6 h)/R(12 h); 3) kae + H/R group, pretreatment with kaempferol (20 $\mu$M) for 2 h prior to H (6 h)/R (12 h); 4) Kae + H/R + NC, transfection with negative control (NC) followed by treatment with kaempferol (20 $\mu$M) for 2 h and then co-treatment with H (6 h)/R (12 h); 5) Kae + H/R + miR-21 I, transfection with miR-21 inhibitor (miR-21 I) followed by treatment with kaempferol (20 $\mu$M) for 2 h and then co-treatment with H (6 h)/R (12 h); and 6) miR-21 I alone group. To demonstrate the role of Notch1/PTEN/Akt signaling pathway in the cardioprotection of kaempferol, cells were randomly allocated to five groups:1) Control group, no treatment; 2) H/R group, H (6 h)/R(12 h); 3) Kae + H/R group, pretreatment with kaempferol (20$\mu$M) for 2 h prior to H (6 h)/R (12 h); 4) Kae + H/R + Control siRNA, transfection with control siRNA followed by treatment with kaempferol (20 $\mu$M) for 2 h and then co-treatment with H (6 h)/R (12 h); and 5) Kae + H/R + Notch1 siRNA, transfection with Notch1 siRNA followed by treatment with kaempferol (20 $\mu$M) for 2 h and then co-treatment with H (6 h)/R (12 h).

### Cell Counting Kit (CCK)-8 assay

The cell viability was analyzed by a CCK-8 assay kit (Dojindo Molecular Technologies, Inc., Kumamoto, Japan) according to a standardized method. Briefly, after the corresponding

administration, cells were incubated with 10 μL of CCK-8 solution at 37˚C for 3 h. The absorbance at 450 nm was measured using a microplate reader (Bio-Rad Benchmark, Hercules, CA, USA).

## Lactate dehydrogenase (LDH) assay

Cell damage was also reflected by the measurement of LDH activity in the culture supernatant using an LDH detection kit (Shanghai Genmed Pharmaceutical Technology Co Ltd., Shanghai, China) according to the manufacturer's protocol. In brief, the culture medium was collected and the adherent H9c2 cells were lysed according to the instructions. LDH activity was assessed using a microplate reader (Biotek Corporation, VT, USA) at 490 nm.

## Cell transfection

MiR-21 inhibitor and negative control were synthesized by Biomics Biotechnologies (Nantong, China). Control small interfering RNA (siRNA) and Notch1 siRNA were purchased from Santa Cruz Biotechnology (Santa Cruz, CA, U.S.A.). The sequences were as following: miR-21 inhibitor sense 5′−UAGCUUAUCAGACUGAUGUUGA−3′ and anti-sense 5′−UCAACA UCAGUCUGAUAAGCUA−3′, negative control sense 5′−UUCUCCGAACGUGUCACGUTT−3′ and anti-sense 5′−ACGUGACACGUUCGGAGAATT−3′, Control siRNA forward 5′−CCUAC GCCACCAAUUUCGU−3′ and anti-sense 5′−ACGAAAUUGGUGGCGUAGG−3′, and Notch1-siRNA sense 5′−GCACGCGGAUUAAUUUGCA−3′ and anti-sense 5′−UGCAAAUUAAUCCG CGUGC−3′. H9c2 cells were plated to reach 70–90% confluency and then were transfected with miR-21 inhibitor, negative control, Control siRNA or Notch1 siRNA at a final concentration of 100 nM using Lipofectamine 3000 (Life technologies corporation, Gaithersburg, MD, USA) in Opti-MEM medium (Gibco, Carlsbad, CA, USA) according to the manufacturer's instructions. Transfected cells were subjected to H/R after 48 h of transfection. The efficiency of transfection was determined using reverse transcription-quantitative polymerase chain reaction (RT-qPCR).

## RT-qPCR assay

Total RNA was extracted from the cells with the TRIzol® reagent (Invitrogen; Thermo Fisher Scientific, Inc., Waltham, MA, USA) according to the manufacturer's protocol. For mRNA detection, RNA (1 μg) was reversed transcribed to cDNA using the RevertAid™ cDNA Synthesis kit (Thermo Fisher Scientific, Inc.) according to the manufacturer's instructions. Real-time PCR amplification was performed on a CFX96 RT-PCR system (Biorad, Hemel Hempstead, UK) using iTaq™ Universal SYBR® Green Supermix (Bio-Rad, USA) according to manufacturer's specification. Reaction conditions were as follows: pre-denaturation at 96˚C for 5 min, followed by 45 cycles of denaturation at 95˚C for 30 s and annealing at 56˚C for 45 s, and a final extension at 72˚C for 60 s. β-actin served as an internal control. For miRNA detection, One-step real-time qPCR was performed, and the expression of miR-21 was detected using the EzOmics miRNA qPCR Detection Primer Set (Biomics, USA) and EzOmics One-Step qPCR Kit (Biomics, USA). PCR was carried out at 42˚C for 45 min; 95˚C for 10 min, followed by 45 cycles of amplification. The primers used synthesized by Invitrogen (Shanghai, China) and as follows: Notch-1, forward: 5′−ATGACTGCCCAGGAAACAAC−3′ and reverse: 5′−GTCCAG CCATTGACACACAC−3′; β-actin, forward: 5′−AGGGAAATCGTGCGTGAC−3′ and reverse, 5′−CGCTCATTGCCGATAGTG−3′; miR-21, forward: 5′−GCACCGTCAAGGCTGAGAAC−3′ and reverse: 5′−CAGCCCATCGACTGGTG−3′; and U6, forward: 5′−CTCGCTTCGGCAGCA CA−3′ and reverse: 5′−AACGCTTCACGAATTTGCGT−3′. U6 was used as an internal control. The relative expression was calculated using the $2^{-\Delta\Delta Ct}$ method [39].

## Caspase-3 activity measurement

Caspase-3 activity was measured in lysates of cells using a FluorAce Apopain Assay kit (BioRad, Hercules, CA, USA) following the manufacturer's instructions. In brief, the lysates of cells were incubated with DEVD-pNA substrate (200 $\mu$M, a substrate of caspase-3) at 37˚C for 4 h. The absorbance at 405 nm was measured by a microplate reader (Biotek Corporation, VT, USA).

## Cell apoptosis assay

Cell apoptosis was determined using an Annexin V-fluorescein isothiocyanate (FITC) apoptosis detection kit (BD Biosciences, San Diego, CA, USA) followed by flow cytometry. Briefly, H9c2 cells were collected by 0.25% trypsin, washed twice with PBS, and centrifuged at 1, 000 × g for 5 min. The cells were then incubated with 5 $\mu$L of Annexin V-FITC and 10 $\mu$L of propidium iodide (PI) for 20 min at room temperature in the dark. Apoptotic rate was analyzed by flow cytometry (FACSCalibur; BD Biosciences) with Cell Quest 3.3 software (BD Biosciences, Franklin Lakes, NJ, USA).

## Endogenous reactive oxygen species (ROS) production measurement

Endogenous ROS production was detected using 2',7'-dichlorofluorescein diacetate (DCFH-DA; Sigma-Aldrich, St Louis, USA) according to the manufacturer's protocol. After treatment with corresponding administration, H9c2 cells were collected and incubated with DCFH-DA (25 $\mu$M) at 37˚C for 20 min in the dark and were washed twice times with PBS. ROS generation was determined using a FACSCalibur flow cytometer with Cell Quest 3.3 software (BD Biosciences, Franklin Lakes, NJ, USA) at an excitation wavelength of 488 nm and an emission wavelength of 525 nm.

## Determination of NO level in the culture supernatant

NO production in culture medium was measured using the Nitrate/Nitrite Assay Kit (Beyotime Institute of Biotechnology, Nanjing, China; cat. number S0021M) according to the instructions. Briefly, cell culture medium (50 $\mu$L) were collected, centrifuged at 1, 000 × g for 10 min and mixed with Griess Reagent I (50 $\mu$L) and Griess Reagent II (50 $\mu$L), both included in the kit, in a 96-well microtiter plate for 10 min at 37˚C. The absorbance at 540 nm was read using a microplate reader (Biotek Corporation, VT, USA).

## Detection of MDA content, SOD and GSH-Px activities

The content of MDA (cat. number A003-1), the activities of SOD (cat. number A001-3) and GSH-Px (cat. number A007-1-1) in treated H9c2 cells were measured by using corresponding commercial kits (Nanjing Jiancheng Bioengineering Institute, Nanjing, China) according to the manufacturer's instructions.

## Enzyme-linked immunosorbent assay

The culture supernatant was collected and centrifuged at 10000 × g at 4˚C for 10 min. The concentrations of cytokines including IL-1β (cat. number H002), IL-6 (cat. number H007), TNF-α (cat. number H052), IL-8 (cat. number H008), and IL-10 (cat. number H009), in the culture supernatant, were detected by a commercial enzyme-linked immunosorbent assay (ELISA) kit (Nanjing Jiancheng Bioengineering Institute) according to the manufacturer's instructions, respectively. Optical density was measured at 450 nm.

## Western blot analysis

After treatment, H9c2 cells were collected and lysed with RIPA lysis buffer (Beyotime Institute of Biotechnology; cat. number P0013B) containing 0.5 mM of phenylmethanesulfonyl fluoride (PMSF) (Beyotime Institute of Biotechnology; cat. number ST506). Protein concentration was measured by BCA protein assay (Thermo Fisher Scientific, Grand Island, NY, USA; cat. number 23227) and equal amounts of protein (30 μg) were fractionated by 12% sodium dodecyl sulfate-polyacrylamide gel and then transferred to polyvinylidene difluoride (PVDF) membrane (Millipore, Billerica, MA, USA; cat. number C2035). After blocked with 5% nonfat milk powder in Tris buffer containing 0.1% Tween-20 (TBST) for 2 h at room temperature, the membranes were incubated with the appropriate following primary antibodies: anti-Bax (dilution of 1: 2000; cat. number #2772, Cell Signaling Technology, MA, USA), anti-Bcl-2 (dilution of 1: 1000; cat. number ab32124, Abcam, Britain), anti-Notch1(dilution of 1: 2000; cat. number #3608, Cell Signaling Technology), anti-PTEN (dilution of 1: 1000; cat. number #9188, Cell Signaling Technology), anti-P-AKT (dilution of 1: 2000; cat. number #4060, Cell Signaling Technology), anti-AKT (dilution of 1: 2000; cat. number #4691, Cell Signaling Technology), and anti-GAPDH (dilution of 1: 2000; cat. number #5174, Cell Signaling Technology) overnight at 4°C. GAPDH was used as an internal control. After washing three times with TBST, the membranes were incubated with horseradish peroxidase (HRP)-conjugated second antibodies (dilution of 1: 2000, cat. number #7074, Cell Signaling Technology) for 2 h at room temperature. Images were visualized with an enhanced BeyoECL Plus reagents (Beyotime Institute of Biotechnology; cat. number P0018M) and analyzed by ImageJ 1.49 (National Institute of Health, Bethesda, MD).

## Statistical analysis

Results are presented as the means ± standard deviation (SD) of at least three independent replicates. Data comparisons between the two groups were analyzed using one-way ANOVA followed by the Tukey multiple comparisons test. When the sample does not meet the normality check, a nonparametric test was performed followed by the Dunnett's T3 test. Data was analyzed by GraphPad Prism 5.0 software (GraphPad Software, Inc., La Jolla, CA, USA). $P < 0.05$ was considered to indicate a statistically significant difference.

## Results

### Kaempferol attenuates hypoxia/reoxygenation (H/R)-induced cardiomyocyte injury and promotes miR-21 expression in H9c2 cells

To determine the protective effect of kaempferol on H/R injury, the functions of kaempferol under H/R condition were conducted by detecting cell viability and LDH activity. As shown in Fig 1, pretreatment with kaempferol (10, 20, and 30 μM) obviously increased cell viability (Fig 1A) and decreased LDH activity compared to H/R group (Fig 1B). Kaempferol (20 μM) significantly led to an increase in cell viability and a decrease in LDH release ($P < 0.01$), and the inhibitory efficiency of kaempferol (30 μM) was not significantly higher than that of kaempferol (20 μM) ($P > 0.05$). Therefore, kaempferol at a concentration of 20 μM were performed in subsequent experiments. To further investigate the protection of kaempferol against H/R injury, the impact of kaempferol on apoptosis were detected by Annexin V-FITC/PI staining (Fig 1C). The results revealed that H/R results in a significant increase in apoptosis rate compared with the control group, while kaempferol pretreatment led to the decrease in apoptosis rate compared with H/R group (Fig 1D). Furthermore, western blot results (Fig 1E) revealed that pretreatment with kaempferol also reverses H/R-induced the increase in the expression of

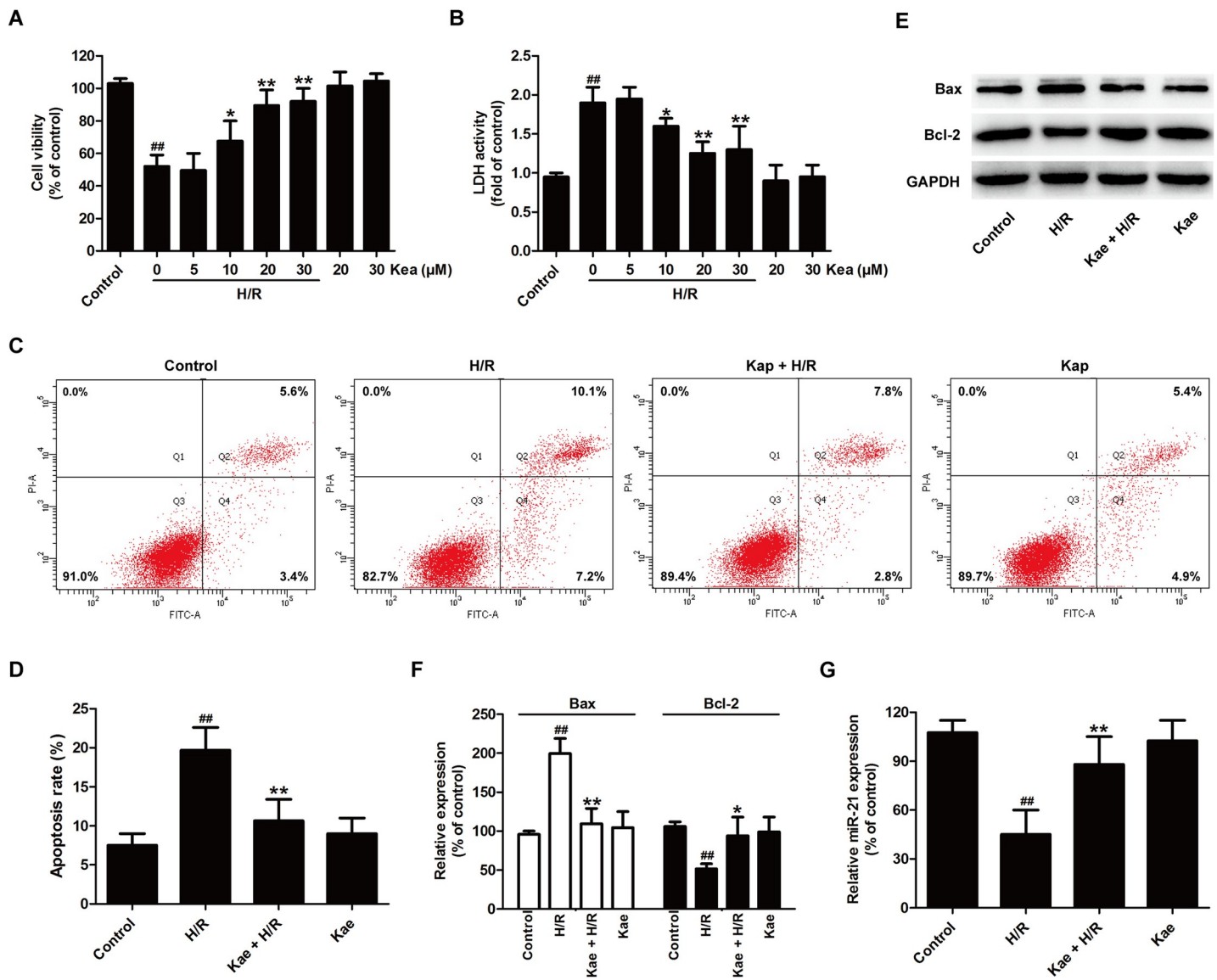

**Fig 1. Kaempferol protects cardiomyocyte injury and increases miR-21 expression in H9c2 cell-subjected with H/R.** H9c2 cells were pretreated with kaempferol (5, 10, 20, and 30 μM) for 2 h and then co-treated with H (6 h)/R (12 h). (A) CCK-8 assay for cell viability. (B) LDH detection kit for LDH activity in cellular supernatant. Data are presented as the means ± SD, n = 3. Compared with the control group, $^{##}P < 0.001$; compared with the H/R group, $^*P < 0.05$, $^{**}P < 0.01$. H9c2 cells were pretreated with kae (20 μM) for 2 h and then co-treated with H (6 h)/R (12 h). (C) Annexin V-FITC staining assay for apoptosis rates. (D) A statistical representation of apoptosis rates. (E) Western blot analysis for Bax and Bcl-2 expression. (F) Quantitative analysis of Bax and Bcl-2 expressions normal to GAPDH. (G) RT-PCR for miR-21 level. Data are presented as the means ± SD, n = 3. Compared with the control group, $^#P < 0.05$, $^{##}P < 0.001$; compared with the H/R group, $^*P < 0.05$, $^{**}P < 0.01$. Kae, kaempferol; H/R, hypoxia/reoxygenation.

pro-apoptotic protein (Bax) and the decrease in the expression of anti-apoptotic protein (Bcl-2) in H9c2 cells (Fig 1F).

Several studies have proved the possible involvement of miR-21 in ischemia-reperfusion (I/R) injury by regulating key signaling elements in cell survival and apoptosis [40]. The present article further found that following kaempferol pretreatment, the expression of miR-21 was obviously increased in comparison with the H/R group (Fig 1G). Taken together, these data indicated that kaempferol prevents H9c2 cells against H/R injury and enhances miR-21 level in H/R-exposed H9c2 cells.

## MiR-21 inhibitor reverses the protective effect of kaempferol on H/R injury in H9c2 cells

To postulate whether miR-21 mediates the cardioprotection of kaempferol against H/R injury, miR-21 inhibitor was utilized in kaempferol functional assays. RT-PCR results showed that miR-21 inhibitor transfection has succeeded in reducing miR-21 expression compared with negative control (NC) transfection (Fig 2A). Beside, miR-21 inhibitor transfection reversed kaempferol-induced the upregulation of cell viability (Fig 2B) and the downregulation of LDH activity (Fig 2C) compared with NC transfection in H/R-treated H9c2 cellsmiR-21 inhibitor Fig. Additionally, miR-21 inhibitor exhibited an increased apoptosis rate when compared with controls NC transfection (Fig 2D) in cotreated with kaempferol and H/R group. Caspase-3 plays a vital function in the intrinsic and extrinsic process of apoptosis [41]. The results further presented that kaempferol inhibits H/R-induced the increase in caspase-3 activity, while this effect was reversed by miR-21 inhibitor transfection (Fig 2E). Furthermore, western blot results (Fig 2F) showed that kaempferol-resulted in a decrease in Bax expression and an increase in Bcl-2 expression in H/R-exposed H9c2 cells are also blocked by miR-21 inhibitor transfection (Fig 2G). These data indicated that miR-21 mediates the protective effect of kaempferol on H/R injury partly by inhibiting mitochondrial-dependent apoptosis in H9c2 cells.

## MiR-21 inhibitor eliminates kaempferol-induced the inhibition on oxidative stress in H/R-treated H9c2 cells

Oxidative stress, defined as an imbalance between production and destruction of free radicals, participates in apoptosis, mitochondrial dysfunction, and the pathogenesis and progression of myocardial I/R injury [4]. Hence, the present further studied whether kaempferol regulates reactive oxygen species (ROS) production and antioxidant defense under H/R condition and whether miR-21 participates in this process. As illustrated in Fig 3, kaempferol mostly reduced the endogenous ROS production (Fig 3A) and malondialdehyde (MDA) content (Fig 3B), the latter of which is a measure of ROS-related lipid peroxidation. Consistently, these effects were abolished by miR-21 inhibitor transfection. In addition, kaempferol markedly enhanced the activities of antioxidant enzymes including superoxide dismutase (SOD) (Fig 3C) and glutathione peroxidase (GSH-Px) (Fig 3D) compared with H/R group in H9c2 cells, which were both reversed bymiR-21 inhibitor transfection. MiR-21 inhibitor alone transfection had no effects on oxidative stress. These results revealed that miR-21 mediates the protection of kaempferol against oxidative injury under the H/R condition in H9c2 cells.

## MiR-21 inhibitor mitigates the protective effect of kaempferol on inflammatory response under H/R injury in H9c2 cells

Considerable evidence has revealed a causative function of inflammatory response in I/R-induced cardiac injury [42]. The present study further demonstrated that kaempferol significantly reduces the concentrations of pro-inflammatory cytokines IL-1β (Fig 4A), IL-8 (Fig 4B) and TNF-α (Fig 4C) and remarkably increases the concentration of anti-inflammatory cytokine IL-10 (Fig 4D) as relative to H/R group. However, the protective effect of kaempferol on inflammatory response was obviously attenuated by miR-21 inhibitor transfection. Nitric oxide synthase (iNOS)/subsequent nitric oxide (NO) pathway is known to be involved in the regulation of many physiological processes, such as inflammation and cardiomyocyte survival [43, 44]. The results further confirmed that H/R-induced the excessive increases in iNOS activity (Fig 4E) and NO level (Fig 4F) in H9c2 cells were attenuated by kaempferol. However,

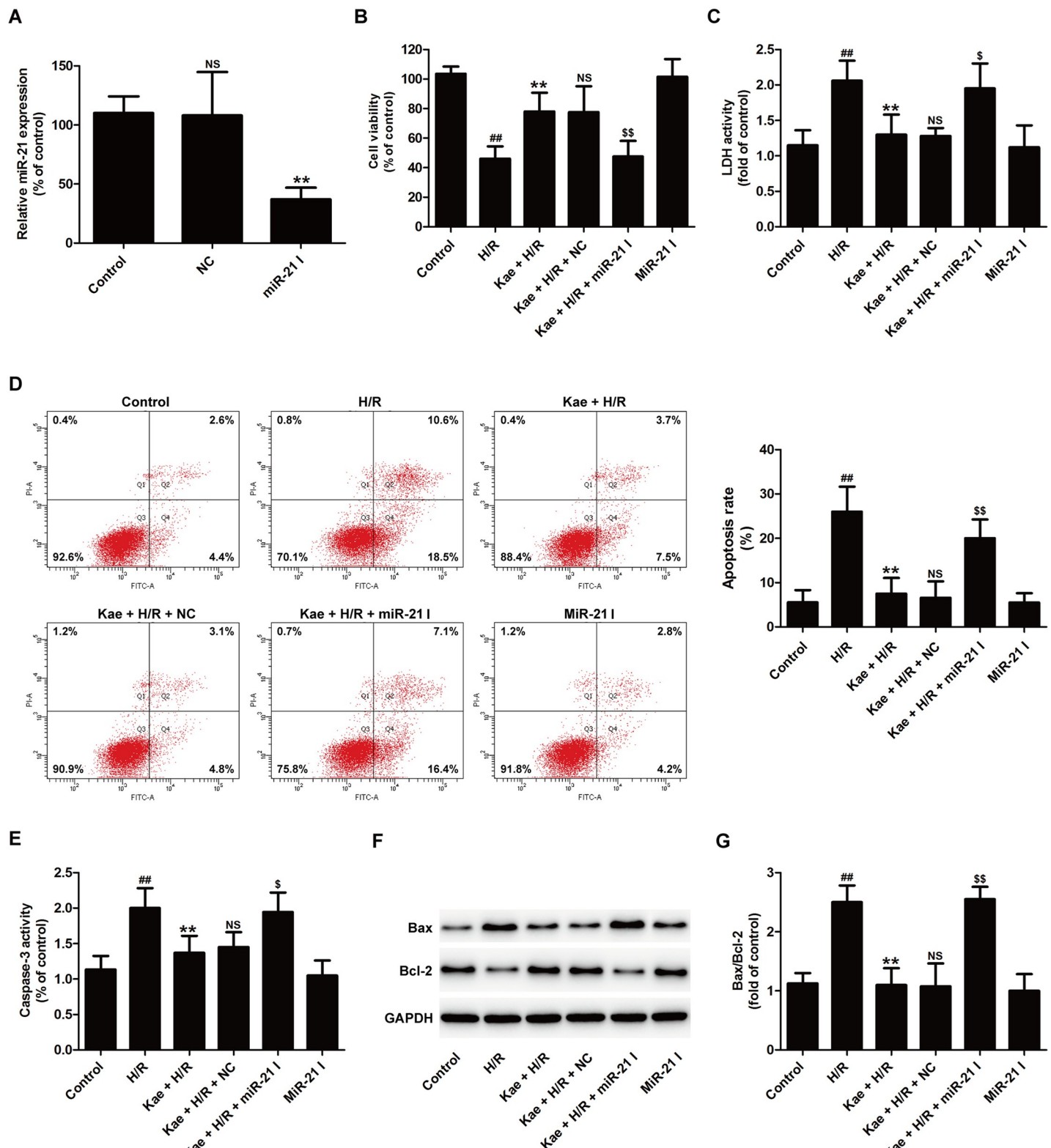

**Fig 2. MiR-21 inhibitor attenuates kaempferol-induced the inhibitory effects on H/R-induced cytotoxicity and apoptosis in H9c2 cells.** H9c2 cells were transfected with miR-21 inhibitor (miR-21 I) or negative control (NC). (A) RT-PCR for miR-21 level. Data are presented as the means ± SD, n = 3. Compared with the control group, $^{NS}P > 0.05$; compared with the NC transfection group, $^{**}P < 0.01$. H9c2 cells were transfected with miR-21 I or NC followed by treatment with kaempferol (20 μM) for 2 h and then co-treatment with H (6 h)/R (12 h). (B) CCK-8 assay for cell viability. (C) LDH detection kit for LDH activity in cellular supernatant. (D) Annexin

V-FITC staining assay for apoptosis. (E) Determination of caspase-3 activity by commercial kit. (F) Western blot analysis for Bax and Bcl-2 expression. (G) Quantitative analysis of Bax/Bcl-2 normal to GAPDH. Data are presented as the means ± SD, n = 3. Compared with the control group, $^{\#}P < 0.05$, $^{\#\#}P < 0.001$; compared with the H/R group, $^{*}P < 0.05$, $^{**}P < 0.01$; compared with the Kae + H/R group, $^{NS}P > 0.05$; compared with the Kae + H/R + NC group, $^{\$}P < 0.05$, $^{\$\$}P < 0.01$. NS: No significance. Kae, kaempferol; H/R, hypoxia/reoxygenation.

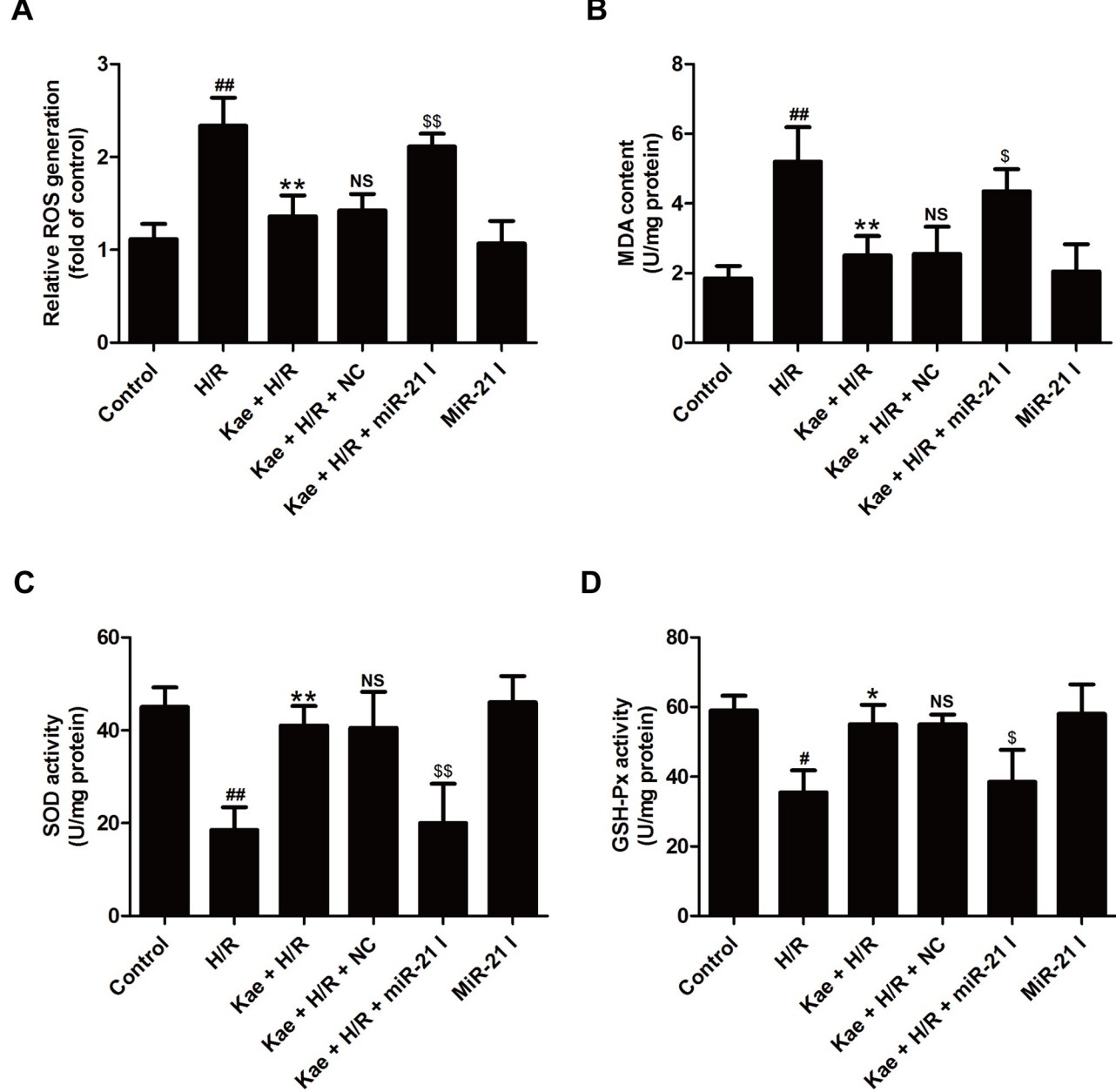

**Fig 3. MiR-21 inhibitor reverses kaempferol-induced inhibition on oxidative stress in H9c2 cells exposed to H/R.** H9c2 cells were transfected with miR-21 inhibitor (miR-21 I) or negative control (NC) followed by treatment with kaempferol (20 $\mu$M) for 2 h and then co-treatment with H (6 h)/R (12 h). (A) ROS production was detected by 2',7'-dichlorofluorescein diacetate (DCFH-DA). (B) Malondialdehyde (MDA) content, (C) superoxide dismutase (SOD), and (D) glutathione peroxidase (GSH-Px) were measured by corresponding commercial kits. Data are presented as the means ± SD, n = 3. Compared with the control group, $^{\#}P < 0.05$, $^{\#\#}P < 0.001$; compared with the H/R group, $^{*}P < 0.05$, $^{**}P < 0.01$; compared with the Kae + H/R group, $^{NS}P > 0.05$; compared with the Kae + H/R + NC group, $^{\$}P < 0.05$, $^{\$\$}P < 0.01$. Kae, kaempferol; H/R, hypoxia/reoxygenation.

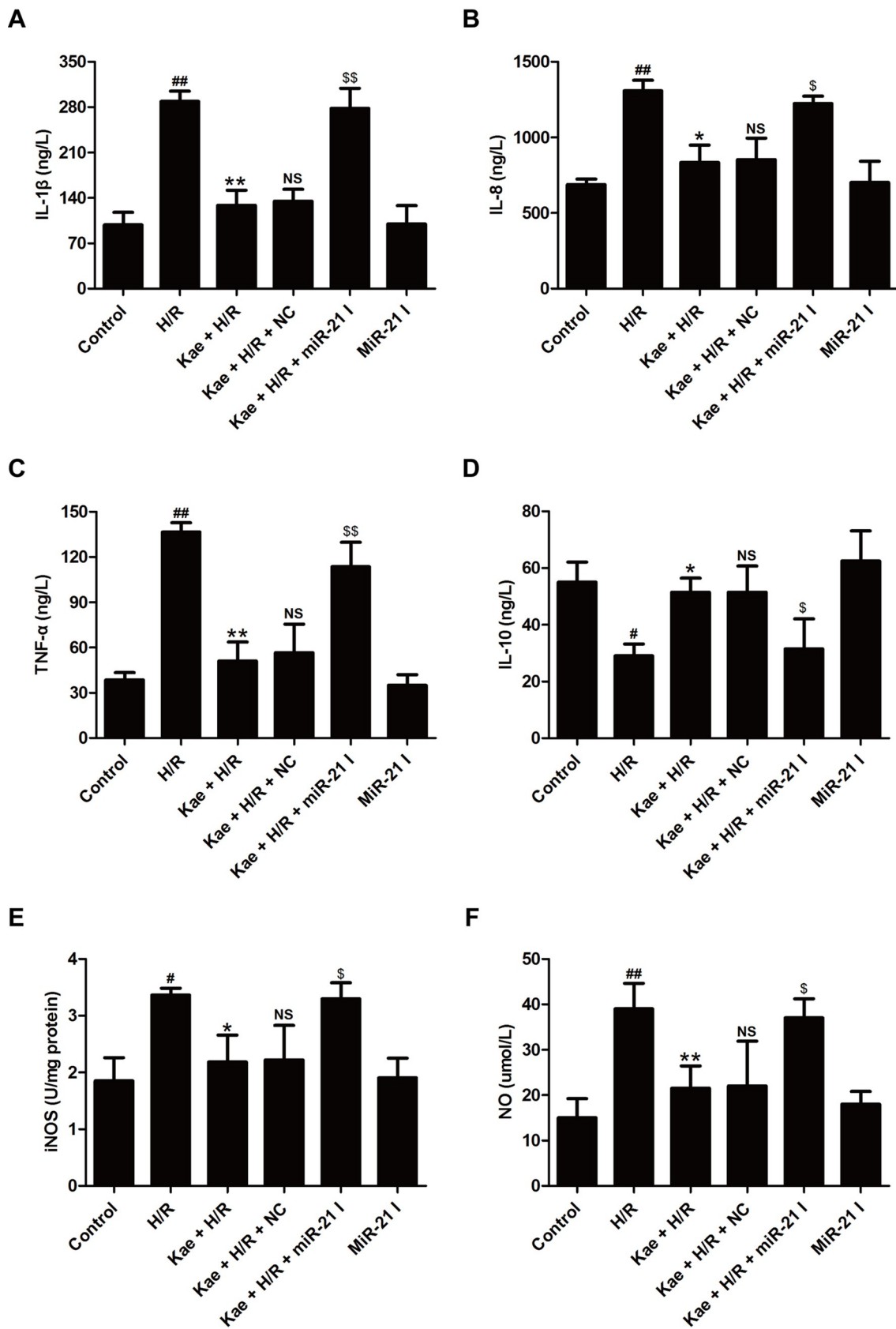

**Fig 4. MiR-21 inhibitor blocks kaempferol-induced the inhibition on inflammatory response in H9c2 cells exposed to H/R.** H9c2 cells were transfected with miR-21 inhibitor (miR-21 I) or negative control (NC) followed by treatment with kaempferol (20 $\mu$M) for 2 h and then co-treatment with H (6 h)/R (12 h). The levels of IL-1β (A), IL-8 (B), TNF-α (C) and IL-10 (D), and the activity of iNOS were determined by ELISA, respectively. (F) The content of NO was analyzed by the Nitrate/Nitrite Assay Kit. Data are presented as the means ± SD, n = 3. Compared with the control group, [#]$P < 0.05$, [##]$P < 0.001$; compared with the H/R group, [*]$P < 0.05$, [**]$P < 0.01$; compared with the Kae + H/R group, [NS]$P > 0.05$; compared with the Kae + H/R + NC group, [$]$P < 0.05$, [$$]$P < 0.01$. Kae, kaempferol; H/R, hypoxia/reoxygenation.

miR-21 inhibitor transfection reversed the inhibition of kaempferol on the iNOS/NO pathway (Fig 4E and 4F). Overall, these results implied that miR-21 mediates kaempferol-alleviated H/R-induced inflammatory response partially by inhibiting iNOS/NO pathway.

## Kaempferol attenuates H/R injury via promoting Notch1/PTEN/Akt signaling pathway in H9c2 cells

Recently, the Notch1 signaling pathway has attracted growing concerns for its myocardial protective actions [33, 34]. To explore whether the Notch1 signaling pathway plays a role in the present circumstance, the effect of kaempferol on Notch 1 expression was investigated. Western blot analysis results (Fig 5A) showed that H/R treatment decreases Notch1 protein expression, which was reversed by kaempferol (Fig 5B). It is reported that phosphatase and tensin homolog deleted on chromosome ten (PTEN) can be negatively regulated by Notch 1 signaling, and is the upstream and negative regulator of AKT pathway [36]. The effects of kaempferol on PTEN/Akt pathway were further investigated. Western blot results showed that kaempferol obviously decreases PTEN expression (Fig 5C) and increases P-Akt/Akt ratio (Fig 5D) compared with H/R group. These results indicated the promotion of kaempferol on the Notch1/PTEN/Akt signaling pathway under the H/R condition.

To confirm the function of the Notch1/PTEN/Akt signaling pathway, the loss of function assay was carried out. Notch1 siRNA transfection significantly reduced the Notch1 protein expression compared with the control siRNA transfection (Fig 5E). In addition, RT-PCR results showed thatNotch1 siRNA transfection also blocked kaempferol-induced an increase in Notch1 miRNA level and a decreases in PTEN miRNA level, and an increase in Akt miRNA level in H/R-treated H9c2 cells (Fig 5F). These data indicated that Notch1 siRNA lead to the blockage of kaempferol-induced the activation of Notch1/PTEN/Akt signaling pathway under H/R condition. On these foundations, results further revealed that Notch1 siRNA attenuates kaempferol-induced the up-regulation of cell viability and the down-regulation of LDH activity in H9c2 cells exposed H/R. All in all, these data implied that Notch1/PTEN/Akt signaling pathway mediates the cardioprotection of kaempferol against H/R injury in H9c2 cells.

## MiR-21 contributes to kaempferol-led to the activation of Notch1/PTEN/Akt signaling pathway in H9c2 cells exposed to H/R

To clarify the further molecular mechanism of miR-21-mediated the beneficial effects of against H/R injury, the effects of miR-21 on the Notch1/PTEN/Akt signaling pathway were investigated. Western blot results (Fig 6A) showed that compared with NC transfection, miR-21 inhibitor transfection reversed kaempferol-induced an increase in Notch1 expression (Fig 6B), a decrease in PTEN (Fig 6C), and an increase in P-Akt/Akt (Fig 6D) in H/R-treated H9c2 cells. MiR-21 inhibitor alone transfection did not affect Notch1/PTEN/Akt signaling pathway. These results suggested that miR-21 mediates kaempferol-induced the enhancement of the Notch1/PTEN/Akt signaling pathway along with myocardial protection.

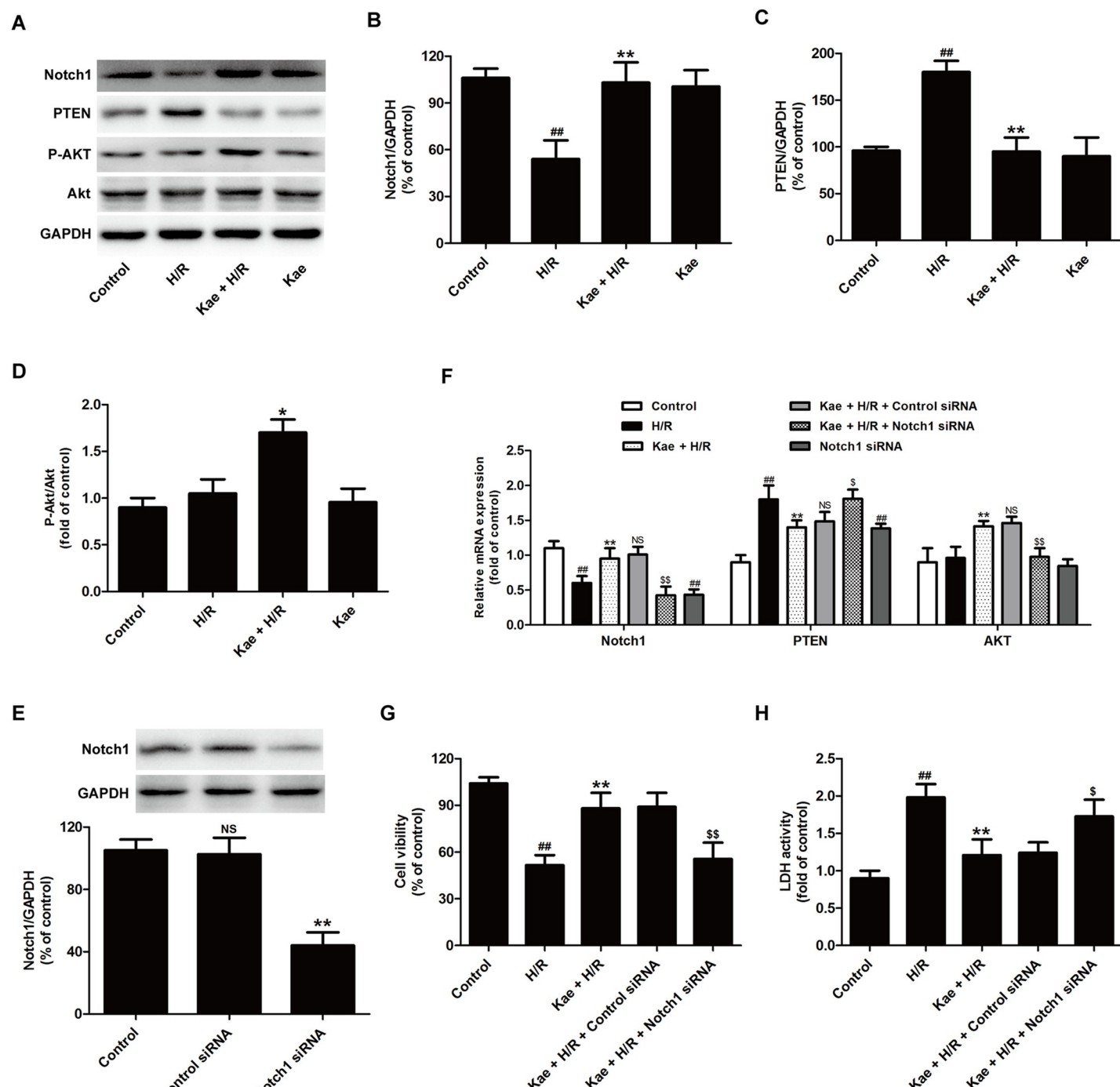

**Fig 5. Kaempferol blocks H/R-induced the inhibition on Notch1/PTEN/Akt signaling pathway in H9c2 cells exposed to H/R.** H9c2 cells were pretreated with kaempferol (20 $\mu$M) for 2 h and then co-treated with H/R. (A) Protein expressions were measured by Western blot analysis. Quantitative analysis of Notch1 (B), PTEN (C), and P-Akt/Akt (D) normal to GAPDH. Data are presented as the means ± SD, n = 3. Compared with the control group, $^{\#\#}P < 0.01$; compared with the H/R group, $^{*}P < 0.05$, $^{**}P < 0.01$. H9c2 cells were transfected with control siRNA and Notch1 siRNA, respectively. (E) Western blot analysis for Notch1 normal to GAPDH. Data are presented as the means ± SD, n = 3. Compared with the control group, $^{NS}P > 0.05$; compared with the control siRNA group, $^{**}P < 0.01$. H9c2 cells were transfected with control siRNA or Notch1 siRNA followed by treatment with kaempferol (20 $\mu$M) for 2 h and then co-treated with H/R. (F) RT-PCR for mRNA level and Quantitative analysis. (G) CCK-8 assay for cell viability. (H) LDH detection kit for LDH activity in cellular supernatant. Data are presented as the means ± SD, n = 3. Compared with the control group, $^{\#\#}P < 0.001$; compared with the H/R group, $^{**}P < 0.01$; compared with the Kae + H/R + control siRNA group, $^{\$}P < 0.05$, $^{\$\$}P < 0.01$. Notch1, neurogenic locus notch homolog protein-1; PTEN, phosphatase and tensin homolog deleted on chromosome ten.

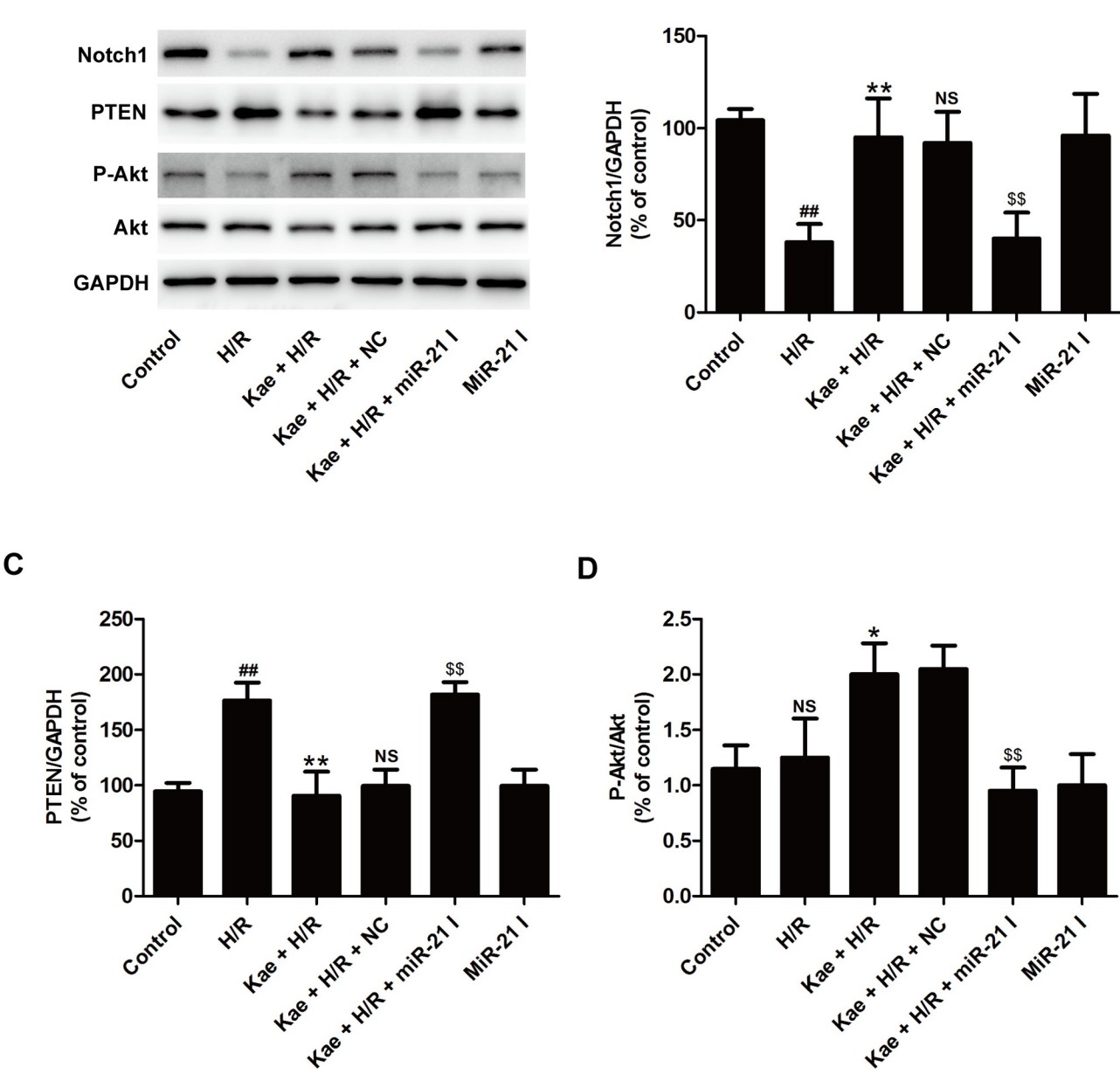

**Fig 6. MiR-21 inhibitor abolishes kaempferol-induced the activation of Notch1/PTEN/Akt signaling pathway in H9c2 cells exposed to H/R.** H9c2 cells were transfected with miR-21 inhibitor (miR-21 I)or negative control (NC) followed by treatment with kaempferol (20 $\mu$M) for 2 h and then co-treatment with H (6 h)/R (12 h). (A) Western blot analysis for protein expression. (B) Quantitative analysis of Notch1 expression (C), PTEN expression, and (D) P-Akt/Akt. Data are presented as the means ± SD, n = 3. Compared with the control group, #$P < 0.05$, ##$P < 0.001$; compared with the H/R group, *$P < 0.05$, **$P < 0.01$; compared with the Kae + H/R group, NS$P > 0.05$; compared with the Kae + H/R + NC group, $$P < 0.05$, $$$P < 0.01$.

## Discussion

In the present study, the underlying protective mechanisms of kaempferol on cardiac cell injury in response to H/R challenge were investigated. We found that miR-21 mediates the profound myocardial protective effect of kaempferol against H/R injury by reducing oxidative stress and informatory response. Additionally, the Notch1/PTEN/Akt signaling pathway contributed to the therapeutic effect of kaempferol on H/R injury. Furthermore,miR-21 was

required for kaempferol-induced the activation of Notch1/PTEN/Akt signaling pathway H9c2 cells exposed to H/R. Taken together, these results identified a specific mechanism by which kaempferol attenuates H/R injury in H9c2 cells via promoting Notch1/PTEN/Akt signaling pathway in miR-21-dependent manner.

Studies have proved that many flavonoids possessed unique antioxidants and protection against myocardial I/R injury [45–47]. Kaempferol, a naturally occurring flavonoid, also has a very good antioxidant, anti-inflammatory, and cardioprotective properties in myocardial I/R injury [11, 12]. Kaempferol mitigates myocardial ischemic injury by attenuating inflammation and apoptosis in rats [12]. The study from Zhen Guo *et al.* also reveals that kaempferol protects cardiomyocytes against anoxia/reoxygenation-induced injury through inhibiting mitochondria-mediated apoptosis pathway [48]. Analogously, the present study confirmed that kaempferol pretreatment attenuates H/R-induced cytotoxicity and apoptosis in H9c2, indicating the cardioprotection of kaempferol against H/R injury. However, the underlying mechanism of kaempferol under H/R injury remains unclear.

MicroRNA-21 (miR-21) is a highly specific miRNA in heart, which can be potential therapeutic targets for the treatment of cardiovascular-related diseases, such as heart failure [49], atherosclerosis [50], and myocardial infarction [51]. More recently, the functions of miR-21 inhibitor in myocardial I/R injury have received significant attention. Y-Q Pan *et al.* report that myocardial I/R injury leads to a significant decreased miR-21 expression, and miR-21 overexpression effectively inhibited myocardial apoptosis and inflammatory factors release in myocardial I/R-administrated rats [26]. Other researches also reveal that miR-21 mediates a variety of myocardial protective drugs against myocardial I/R injury [29, 52]. Interestingly, a study confirms that kaempferol inhibits vascular smooth muscle cell migration by modulating miR-21 expression [53]. However, the role of miR-21 in the protective effects of kaempferol on myocardial I/R injury have not been reported. In the present study, the results firstly revealed that kaempferol pretreatment remarkably increases miR-21 expression during H/R in H9c2 cells. And, loss-of-function experiments confirmed that down-regulation of miR-21 attenuates the protective effects of kaempferol against H/R-induced cytotoxicity and apoptosis. These data indicated that miR-21 mediates the cardioprotection of kaempferol against H/R injury. However, the molecular mechanisms involved in the contribution of miR-21 to the cardioprotection of kaempferol remain elusive.

More and more studies confirm that kaempferol protects against I/R injury by attenuating oxidative stress, inflammation, and apoptosis [12, 13]. Consistent with these studies, our findings showed that kaempferol decreases ROS generation and MDA content and increases SOD and GSH-Px activities in H9c2 cells exposed to H/R. Additionally, kaempferol reduced pro-inflammatory cytokinesIL-1β, IL-8, and TNF-α levels and promoted anti-inflammatory cytokine IL-10 level as well as attenuating iNOS activity and NO level under H/R condition. These results indicated that kaempferol elicited cardioprotection against I/R injury depending on it's anti-oxidant and anti-inflammatory activities. Hai-Xiang Xu *et al.* prove that miR-21 inhibitor exacerbates oxidative stress and attenuates the antioxidative defense system induced by H/R [27]. Another study reveals that miR-21 mediates the protective effects of hydrogen sulfide against ischemia injury through attenuating inflammatory injury in cardiomyocytes [54]. Consistent with these studies, the present showed that miR-21 inhibitor eliminates kaempferol-induced anti-oxidant and anti-inflammatory activities in H/R-treated H9c2 cells. These results suggested that miR-21 contributes to the anti-oxidant and anti-inflammatory activities of kaempferol during H/R injury.

The Notch1 signaling pathway regulates a wide range of physiological processes, including cell proliferation, apoptosis, transcriptional regulation, and ROS generation, playing important regulatory functions during I/R process [33–35]. Previous studies reveal that pharmacological

activation of the Notch1 signaling pathway attenuated myocardial oxidative stress and inflammation and preserved heart function during I/R injury [55, 56], and knockdown of Notch1 exacerbated cardiac damage following I/R damage by promoting oxidative stress [57]. On the other hand, Liming Yu *et al*. prove that Notch1/Hairy and enhancer of split (Hes)-mediated activation of phosphatase and tensin homolog (PTEN)/Akt signaling plays a crucial role in polydatin-induced protection against I/R injury by ameliorating oxidative/nitrative stress damage [35]. However, there are few studies on the role of the Notch1 signaling pathway in the biological function of kaempferol. Our results showed that kaempferol promotes Notch1 expression, inhibits PTEN expression, and enhances Akt phosphorylation in H9c2 cells exposed to H/R, indicating the activation of Notch1/PTEN/Akt signaling pathway induced by kaempferol during H/R. Besides, knockdown of Notch1 induced by Notch1 siRNA reversed the protection of kaempferol against H/R injury. These results indicated that promoting Notch1/PTEN/Akt signaling pathway contributes to the cardioprotection of kaempferol against H/R injury.

The research draws out the crosstalk between miR-21 and the Notch1/PTEN pathway in Human Colorectal Cancer [58]. Jian Cao *et al*. demonstrates that miR-21 cooperate to modulate the proliferation of vascular smooth muscle cell via regulating Notch2signaling pathway [37]. Notably, during I/R and H/R, forced miR-21 expression promotes Akt signaling activity via suppressing PTEN expression and further inhibited apoptosis and cardiocyte injury [59]. However, the effects of miR-21 on the Notch1/PTEN/Akt pathway under the cardioprotection of kaempferol have not been explored. In the present study, our results further found that miR-21 inhibitor reverses kaempferol-induced the activation of Notch1/PTEN/Akt signaling pathway under H/R condition in H9c2 cells, indicating that kaempferol protects H9c2 cells against H/R injury by promoting Notch1/PTEN/Akt signaling pathway in a miR-21-dependent manner.

However, there is still more to explore behind these experimental results. Firstly, our study is limited to *in vitro* cell experiment, it is of great necessity to perform the animal investigation. Secondly, other study demonstrates that the Notch pathway controls miR-21 expression. During myocardial I/R injury, the possible interplay between miR-21 and the Notch1 signaling pathway has not been further investigated. Thirdly, how miR-21 participates in the regulation of kaempferol on Notch1/PTEN/Akt signaling pathway during myocardial I/R injury remains to be further studied.

Taken together, the results showed that miR-21 mediates the protection of kaempferol against H/R injury by inhibiting oxidative stress and inflammation via promoting Notch1/PTEN/Akt signaling pathway in cardiomyocytes.

## Supporting information

**S1 File.**
(DOC)

## Author Contributions

**Data curation:** Zhenhui Qi.

**Funding acquisition:** Jinxi Huang.

**Investigation:** Zhenhui Qi.

**Project administration:** Jinxi Huang, Zhenhui Qi.

**Writing – original draft:** Zhenhui Qi.

**Writing – review & editing:** Jinxi Huang.

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
