## [Decision Letter · Decision Letter 0]

5 Aug 2020

PONE-D-20-20790

MiR-21 mediates the protection of kaempferol against hypoxia/reoxygenation injury via modulating Notch1/PTEN/AKT signaling pathways in H9c2 cells

PLOS ONE

Dear Dr. Huang,

Thank you for submitting your manuscript to PLOS ONE. After careful consideration, we feel that it has merit but does not fully meet PLOS ONE’s publication criteria as it currently stands. Therefore, we invite you to submit a revised version of the manuscript that addresses the points raised during the review process.

Please adequately address reviewers's concerns.

We look forward to receiving your revised manuscript.

Kind regards,

Meijing Wang, MD

Academic Editor

PLOS ONE

Journal Requirements:

2.PLOS ONE now requires that authors provide the original uncropped and unadjusted images underlying all blot or gel results reported in a submission’s figures or Supporting Information files. This policy and the journal’s other requirements for blot/gel reporting and figure preparation are described in detail at https://journals.plos.org/plosone/s/figures#loc-blot-and-gel-reporting-requirements and https://journals.plos.org/plosone/s/figures#loc-preparing-figures-from-image-files. When you submit your revised manuscript, please ensure that your figures adhere fully to these guidelines and provide the original underlying images for all blot or gel data reported in your submission. See the following link for instructions on providing the original image data: https://journals.plos.org/plosone/s/figures#loc-original-images-for-blots-and-gels.

Reviewers' comments:

Reviewer's Responses to Questions

**Comments to the Author**

1. Is the manuscript technically sound, and do the data support the conclusions?

Reviewer #1: Yes

Reviewer #2: Yes

2. Has the statistical analysis been performed appropriately and rigorously? 

Reviewer #1: Yes

Reviewer #2: Yes

3. Have the authors made all data underlying the findings in their manuscript fully available?

Reviewer #1: Yes

Reviewer #2: No

4. Is the manuscript presented in an intelligible fashion and written in standard English?

Reviewer #1: No

Reviewer #2: No

5. Review Comments to the Author

Reviewer #1: Kaempferide is a nature flavanol. Previously, other reports demonstrated that kaempferol improves I/R-induced cardiac dysfunction and myocardial injury via reducing myocardial infarct size, cardiomyocyte apoptosis, oxidative stress. In this study, Huang and Qi reported that kaempferol mediated protective effects in H9c2 cells against hypoxia/reoxygenation injury could be mediated by modulating of miR-21 expression to regulate Notch1/PTEN/AKT signaling pathways. The overall of the study is well designed and executed. Some major and minor issued need to be addressed are as below:

1. I/R injury induced cardiomyocyte drop-out is mostly associated with necrosis rather than in vitro H/R cell model associated cell apoptosis. Thus, using H9c2 cells in H/R model in this study may not fully reflect the true mechanism of kaempferide mediated in vivo cardiac-protective effects again I/R injury.

2. In Figure1 and Figure 2 D, the Flow cytometry analysis results of apoptosis (Fitc-Annexin positive) , late stage apoptosis (FITC-A and PI double positive) and necrosis (PI positive) is not consistent.

3. This study only examined overall Notch-1 transcription and protein level, but did not check the Notch intracellular domain (NICD). As NICD is a more direct reflection of Notch signaling activation, so it should be included in the WB analysis.

4. In methods section at Page 10, some of the words are written in Chinses which need to be replaced by English.

Reviewer #2: Ref: PONE-D-20-20790

In the present article entitled “MiR-21 mediates the protection of kaempferol against hypoxia/ reoxygenation injury via modulating Notch1/PTEN/AKT signaling pathways in H9c2 cells” Huang et al. have explored the involvement of miR-21 in the cardioprotective effect of kaempferol on hypoxia/reoxygenation (H/R)-induced H9c2 cell injury. Further they have tested the involvement of Notch/PTEN/Akt signaling pathway in such cardioprotective effect of kaempferol. The study design is simple and the results are well supported by the data. However, some points need to be addressed to make the study robust for the publication.

1. There are many grammatical errors in the manuscript. Especially the use of “the” needs to be checked throughout the manuscript.

2. Title is not clear. “protection of kaempferol against hypoxia/reoxygenation injury…” needs to be rephrased.

3. Authors have used “Kaempferide” or “kaempferol” or “kae” interchangeably. Please stick with one nomenclature.

4. Page 11, line 11 “Particularly, miR-21 is highly expressed in cardiomyocytes and has also been shown to regulate important processes of myocardial I/R injury”. Please provide some mechanistic information of how miR-21 is involved in myocardial I/R injury by citing other papers like Roy et al., PMID: 19147652; Gu et al., PMID: 25809568; Xu et al., PMID: 25159851.

5. Page 16, line 7, there is some text in different language. Please fix.

6. Statistical analysis. The samples size used in this paper is majorly 3. In this smaller number of samples, the normality check is not possible. Have the authors used non-parametric tests? Please mention.

7. Figure 1, Authors have selected 20 μM concentration of kae for all experiments. But in figure 1A and 1B, the concentration of kea used was 30 μM (without H/R, last bar). Authors should present data from 20 μM (without H/R) in some experiments.

8. Figure 1D: The bar graph (H/R group) presented does not match with the scatter plot data shown in Figure 1C.

9. Page 20, line 10, Figures 1F and 1G are not cited.

10. Page 20, line 13, “and apoptosis [37]. The present further found that following kae pretreatment”. It should be “The present article further…...”

11. Figure 2D: The bar graph (H/R and Kae+H/R+miR-21 I groups) presented does not match with the scatter plot data.

6. PLOS authors have the option to publish the peer review history of their article (what does this mean?). If published, this will include your full peer review and any attached files.

Reviewer #1: No

Reviewer #2: No

---

## [Author Response · Author response to Decision Letter 0]

21 Sep 2020

Dear Editors and Reviewers:

Thank you for your letter and for the comments concerning our manuscript entitled “MiR-21 mediates the protection of kaempferol against hypoxia/reoxygenation injury via modulating Notch1/PTEN/AKT signaling pathways in H9c2 cells (ID: PONE-D-20-20790)”. Those comments are all valuable and very helpful for revising and improving our paper, as well as the important guiding significance to our researches. We have studied comments carefully and have made correction which we hope meet with approval. Revised portion are marked in red in the paper. The main corrections in the paper and the responds to the reviewer’s comments are as following: 

Journal Requirements:

Response: The manuscript has been corrected according to the PLOS ONE's style requirements.

Response: The original uncropped and unadjusted images underlying all blot or gel results are all reported in a Supporting Information files “Full-length blots are presented in Supplementary Figures”.

Response: The ORCID iD has been added.

Review Comments to the Author

Reviewer #1: Kaempferide is a nature flavanol. Previously, other reports demonstrated that kaempferol improves I/R-induced cardiac dysfunction and myocardial injury via reducing myocardial infarct size, cardiomyocyte apoptosis, oxidative stress. In this study, Huang and Qi reported that kaempferol mediated protective effects in H9c2 cells against hypoxia/reoxygenation injury could be mediated by modulating of miR-21 expression to regulate Notch1/PTEN/AKT signaling pathways. The overall of the study is well designed and executed. Some major and minor issued need to be addressed are as below:

1. I/R injury induced cardiomyocyte drop-out is mostly associated with necrosis rather than in vitro H/R cell model associated cell apoptosis. Thus, using H9c2 cells in H/R model in this study may not fully reflect the true mechanism of kaempferide mediated in vivo cardiac-protective effects again I/R injury.

Response: Thanks for your constructive suggestion. We don't know if the reviewers have misunderstanding. A wide array of studies confirms that I/R injury induced cardiomyocyte drop-out is mostly associated with in vitro H/R cell model associated cell apoptosis (SEEN: 1. Dang X, Qin Y, Gu C, Sun J, Zhang R, Peng Z. Knockdown of Tripartite Motif 8 Protects H9C2 Cells Against Hypoxia/Reoxygenation-Induced Injury Through the Activation of PI3K/Akt Signaling Pathway. Cell Transplant. 2020;29:963689720949247; 2. Li Y, Liu X. Novel insights into the role of mitochondrial fusion and fission in cardiomyocyte apoptosis induced by ischemia/reperfusion. J Cell Physiol. 2018;233(8):5589-5597; 3. Badalzadeh R, Mokhtari B, Yavari R. Contribution of apoptosis in myocardial reperfusion injury and loss of cardioprotection in diabetes mellitus. J Physiol Sci. 2015 May;65(3):201-1.). In the present study, we only used H9c2 cell to explore the protective mechanism of kaempferide on myocardial I/R injury. Of course, in vitro experiments will make our article more convincing, which is our next experimental plan. This part has been explained in the penultimate paragraph of our discussion.

2. In Figure1 and Figure 2 D, the Flow cytometry analysis results of apoptosis (Fitc-Annexin positive) , late stage apoptosis (FITC-A and PI double positive) and necrosis (PI positive) is not consistent.

Response: These are two separate sets of experiments. The apoptotic rate of Figure 1D and Figure 2D was not consistent, which may be related to cell state, experimental operation process and drug storage time and so on.

3. This study only examined overall Notch-1 transcription and protein level, but did not check the Notch intracellular domain (NICD). As NICD is a more direct reflection of Notch signaling activation, so it should be included in the WB analysis.

Response: Thanks for your meaningful advice. We are sorry we didn't pay attention to this part. Due to the limitation of time and antibody as well as the impact of the COVID-19 pandemic, we can’t make up the experiment for the time being. We have looked at a lot of other relevant literature, which also only detected the expression of Notch1 (SEEN: 1. Solanki A, Yánez DC, Lau CI, Rowell J, Barbarulo A, Ross S, Sahni H, Crompton T. The transcriptional repressor Bcl6 promotes pre-TCR induced differentiation to CD4+CD8+ thymocyte and attenuates Notch1 activation. Development. 2020:dev.192203; 2. Thabassum Akhtar Iqbal S, Tirupathi Pichiah PB, Raja S, Arunachalam S. Paeonol Reverses Adriamycin Induced Cardiac Pathological Remodeling through Notch1 Signaling Reactivation in H9c2 Cells and Adult Zebrafish Heart. Chem Res Toxicol. 2020; 33(2):312-323; 3. Zheng J, Li J, Kou B, Yi Q, Shi T. MicroRNA-30e protects the heart against ischemia and reperfusion injury through autophagy and the Notch1/Hes1/Akt signaling pathway. Int J Mol Med. 2018; 41(6):3221-3230.). In present study, we detected the changes in downstream targets PTEN/AKT signaling of Notch1 pathway, which can indirectly reflect the activation of Noctch1 pathway. On this basis, the loss of NICD expression does not affect our conclusion. Of course, in future studies, we will supplement and further explore the role of Notch1/PTEN/AKT signaling pathway in the protective effects of kaempferol on myocardial ischemia/reperfusion injury.

4. In methods section at Page 10, some of the words are written in Chinses which need to be replaced by English.

Response: Thanks for the editor’s careful suggestion. “少 了 反 向 序 列” has been replaced by “and reverse: 5’-GTCCAGCCATTGACACACAC-3’” in the reversed manuscript.

Reviewer #2: Ref: PONE-D-20-20790

In the present article entitled “MiR-21 mediates the protection of kaempferol against hypoxia/ reoxygenation injury via modulating Notch1/PTEN/AKT signaling pathways in H9c2 cells” Huang et al. have explored the involvement of miR-21 in the cardioprotective effect of kaempferol on hypoxia/reoxygenation (H/R)-induced H9c2 cell injury. Further they have tested the involvement of Notch/PTEN/Akt signaling pathway in such cardioprotective effect of kaempferol. The study design is simple and the results are well supported by the data. However, some points need to be addressed to make the study robust for the publication.

1. There are many grammatical errors in the manuscript. Especially the use of “the” needs to be checked throughout the manuscript.

Response: Thanks for the editor’s good evaluation and kind suggestion. We are really sorry for the typography, format, and layout that are a bit non-standard, and the spelling and syntax errors in our manuscript. We have carefully examined the format and grammar, and corrected errors and deficiencies in articles. Meanwhile, we also invited an English professional and experienced colleagues to improve the scientific writing in our manuscript. The use of “the” has been checked throughout the manuscript. The “unnecessary the” has been removed in the reversed manuscript.

2. Title is not clear. “protection of kaempferol against hypoxia/reoxygenation injury…” needs to be rephrased.

Response: “protection of kaempferol against hypoxia/reoxygenation injury” has been changed to “protective effect of kaempferol on hypoxia/reoxygenation injury”.

3. Authors have used “Kaempferide” or “kaempferol” or “kae” interchangeably. Please stick with one nomenclature.

Response: “Kaempferide” and “kae” have all been changed to “kaempferol”.

4. Page 11, line 11 “Particularly, miR-21 is highly expressed in cardiomyocytes and has also been shown to regulate important processes of myocardial I/R injury”. Please provide some mechanistic information of how miR-21 is involved in myocardial I/R injury by citing other papers like Roy et al., PMID: 19147652; Gu et al., PMID: 25809568; Xu et al., PMID: 25159851.

Response: These three references are inserted into the reversed manuscript as references 22, 23 and 24.

5. Page 16, line 7, there is some text in different language. Please fix.

Response: Thanks for the editor’s careful suggestion. “少 了 反 向 序 列” has been replaced by “and reverse: 5’-GTCCAGCCATTGACACACAC-3’” in the reversed manuscript.

6. Statistical analysis. The samples size used in this paper is majorly 3. In this smaller number of samples, the normality check is not possible. Have the authors used non-parametric tests? Please mention.

Response: The related part “When the sample does not meet the normality check, a nonparametric test was performed followed by the Dunnett's T3 test.” has been added to the “Statistical analysis”.

7. Figure 1, Authors have selected 20 μM concentration of kae for all experiments. But in figure 1A and 1B, the concentration of kea used was 30 μM (without H/R, last bar). Authors should present data from 20 μM (without H/R) in some experiments.

Response: The effects of kea (5, 10, 20, and 30 μM) alone on cell viability and LDH activity have all been detected. Considering the beauty of the drawing, we only added the kea (30 μM) alone group. Taking into account the reviewers' comments, the effects of kea (20) alone on cell viability and LDH activity have been added to Figure 1A and Figure 1B.

8. Figure 1D: The bar graph (H/R group) presented does not match with the scatter plot data shown in Figure 1C.

Response: The mistake in control group and Kea alone group in Figure 1D have been corrected.

9. Page 20, line 10, Figures 1F and 1G are not cited.

Response: The Figures 1F and 1G have been added to Result (Kaempferol (Kae) attenuates the damage of cardiomyocytes and promotes miR-21 expression in hypoxia/reoxygenation (H/R)-treated H9c2 cell).

10. Page 20, line 13, “and apoptosis [37]. The present further found that following kae pretreatment”. It should be “The present article further…...”

Response: “The present further found” has been corrected to “The present article further found”. 

11. Figure 2D: The bar graph (H/R and Kae+H/R+miR-21 I groups) presented does not match with the scatter plot data.

Response: Our statistical results are based on three independent experiments. Hence, sometimes statistical images may be slightly different from typical images. In Figure 2D, there is nothing wrong between the bar graph and the scatter plot data.

---

## [Decision Letter · Decision Letter 1]

7 Oct 2020

MiR-21 mediates the protection of kaempferol against hypoxia/reoxygenation-induced cardiomyocyte injury via promoting Notch1/PTEN/AKT signaling pathway

PONE-D-20-20790R1

Dear Dr. Huang,

We’re pleased to inform you that your manuscript has been judged scientifically suitable for publication and will be formally accepted for publication once it meets all outstanding technical requirements.

Kind regards,

Meijing Wang, MD

Academic Editor

PLOS ONE

Additional Editor Comments (optional):

Reviewers' comments:

Reviewer's Responses to Questions

**Comments to the Author**

1. If the authors have adequately addressed your comments raised in a previous round of review and you feel that this manuscript is now acceptable for publication, you may indicate that here to bypass the “Comments to the Author” section, enter your conflict of interest statement in the “Confidential to Editor” section, and submit your "Accept" recommendation.

Reviewer #1: All comments have been addressed

Reviewer #2: All comments have been addressed

2. Is the manuscript technically sound, and do the data support the conclusions?

Reviewer #1: Yes

Reviewer #2: Yes

3. Has the statistical analysis been performed appropriately and rigorously? 

Reviewer #1: Yes

Reviewer #2: Yes

4. Have the authors made all data underlying the findings in their manuscript fully available?

Reviewer #1: Yes

Reviewer #2: Yes

5. Is the manuscript presented in an intelligible fashion and written in standard English?

Reviewer #1: Yes

Reviewer #2: Yes

6. Review Comments to the Author

Reviewer #1: (No Response)

Reviewer #2: (No Response)

7. PLOS authors have the option to publish the peer review history of their article (what does this mean?). If published, this will include your full peer review and any attached files.

Reviewer #1: No

Reviewer #2: No

---

## [Editor Report · Acceptance letter]

27 Oct 2020

PONE-D-20-20790R1 

MiR-21 mediates the protection of kaempferol against hypoxia/reoxygenation-induced cardiomyocyte injury via promoting Notch1/PTEN/AKT signaling pathway

Dear Dr. Huang:

I'm pleased to inform you that your manuscript has been deemed suitable for publication in PLOS ONE. Congratulations! Your manuscript is now with our production department. 

Kind regards, 

on behalf of

Dr. Meijing Wang 

Academic Editor

PLOS ONE